# Blood pressure and heart rate variability responses following an acute bout of vinyasa yoga and a prolonged seated control: A randomized crossover trial

**Alexis Thrower**[1]*, **Bethany Barone Gibbs**[2], **Abdullah Alansare**[3], **Sally Sherman**[4], **Kelliann Davis**[4]

1 Department of Pathophysiology, Rehabilitation, and Performance, West Virginia University, Morgantown, West Virginia, United States of America, 2 Department of Epidemiology and Biostatistics, West Virginia University, Morgantown, West Virginia, United States of America, 3 Department of Exercise Physiology, King Saud University, Riyadh, Saudi Arabia, 4 Department of Health and Human Development, University of Pittsburgh, Pittsburgh, Pennsylvania, United States of America

* ant00017@mix.wvu.edu

**Data Availability Statement:** Anonymized data is available as S1 Data in this paper.

## Abstract

Vinyasa yoga is moderate-intensity physical activity, yet physiological responses are poorly characterized. The purpose of this study was to examine the effect of a vinyasa yoga session on autonomic/cardiovascular functioning in healthy adults. A randomized crossover design took place at the Physical Activity and Weight Management laboratory (Pittsburgh, PA; n = 18), and included two experimental conditions: 60 minutes of vinyasa yoga or a seated control, and measurements were taken at baseline, 5-minutes, and 65-minute post-conditions. The primary cardiovascular-related outcomes of this study included blood pressure (BP), heart rate (HR), and HR variability (HRV) measures [natural log transformed (ln) standard deviation of normal-to-normal R-R intervals (SDNN), root mean square of successive differences (RMSSD), high frequency (HF), and low frequency to high frequency ratio (LF/HF ratio)]. Linear mixed effects models were used for data analyses. Systolic BP was 8.14 mmHg lower at 5 minutes post yoga (p<0.001) but was not different 65 minutes post, compared to the control. HR was higher at 5- and 65-minutes post yoga compared to the control (10.49/4.70 bpm, respectively, both p<0.01). HRV was lower (worse) at 5 and 65 minutes post for lnSDNN, lnRMSSD, and lnHF (all p<0.01). LF/HF ratio was higher (worse) at 5 minutes post yoga compared to the control (difference = +0.38, p = 0.025), but not different at 65 minutes post between conditions. Compared to prolonged sitting, vinyasa yoga had variable effects on post-session autonomic function including favorable BP responses and unfavorable HR and HRV responses, further investigation is warranted.

## Background

Cardiovascular disease (CVD) is an umbrella term including diseases of the heart and vasculature within the body [1]. The American Heart Association 2022 report found CVD to be the

**Funding:** The author(s) received no specific funding for this work.

**Competing interests:** The authors have declared that no competing interests exist.

leading cause of death worldwide, [2] with previous evidence suggesting that there is a strong inverse relationship between cardiovascular health and all-cause and CVD mortality [3]. CVD is influenced by many health factors and behaviors including smoking, diabetes, overweight/ obesity, poor diet, high cholesterol levels, high blood pressure (BP) and physical inactivity [1, 2] Physical inactivity is prevalent, with 76% of Americans not meeting the 2018 physical activity guidelines; however, meeting these specific activity guidelines can improve one's cardiovascular health [4] Therefore, more Americans should aim to meet the physical activity guidelines to improve cardiovascular health and decrease both all-cause and CVD mortality risk.

Different modes of exercise can be performed to meet current physical activity guidelines like walking, running, or cycling. Millions of Americans are currently practicing yoga, [5] and our group has demonstrated that the vinyasa style of yoga, specifically, is a mode of moderate-intensity physical activity [6]. Yet, unlike more traditional modes of moderate exercise, vinyasa yoga consists of three components including physical postures (asanas) linked together into different series, breathing practices (pranayama), and mindfulness/meditation (dyna). These additional components may result in vinyasa yoga generating similar or more physiological benefits compared to other modes of physical activity [7].

Physical activity bouts of at least a moderate-intensity are known to improve (i.e., lower) post exercise HR [8] and BP [9, 10]. HR and BP changes from different activities are in part due to cardiac autonomic regulation in the body [11]. Heart rate variability (HRV), similar to HR and BP, is another measurement of cardiac autonomic regulation [12] and observing its function may be a powerful tool in understanding CVD risk [13]. HRV is defined as the time variation/fluctuation between consecutive heartbeats occurring due to the influence of the ANS's two branches, the sympathetic and parasympathetic nervous systems [12]. HRV has been found to improve following exercise, but the time point at which HRV increased following exercise varied [11, 14, 15]. Moreover, almost no studies have evaluated HRV responses following a vinyasa yoga bout, despite its moderate intensity level and unique additional components.

To address research gaps about the relationship between vinyasa yoga and ANS function, the purpose of this study was to measure the effect of vinyasa yoga on autonomic regulation of the cardiovascular system (i.e., changes in BP, HR, and HRV) compared to a prolonged seated control condition.

## Methods

### Participants

A convenience sample of 18 unpaid volunteers were recruited for this study between April 1, 2022 through April 25, 2022. Participants were required to be between 18–55 years old and have had any experience with vinyasa yoga to ensure that they could appropriately perform the yoga sequence. Individuals were excluded from this study if they had physical limitations (i.e., musculoskeletal injuries), a medical condition that required medical clearance (i.e., cancer, heart disease, or type 1 or 2 diabetes), presence of a cardiovascular condition (CVD), previous myocardial infarction, or peripheral artery disease), were taking medications that affect HR or BP (i.e., anti-depressants, beta-blockers, etc.), or were pregnant.

Individuals were recruited through flyers distributed to students/staff in the Department of Health and Human Development at the University of Pittsburgh. All interested individuals gave verbal consent to complete a screening session, after which, if found to be eligible, they attended a virtual or in-person consent session to understand study requirements, obtain written consent, and determine vinyasa yoga familiarity. All study procedures were approved by

the University of Pittsburgh Human Research Protection Office (IRB Approval Number: STUDY 22020050).

## Assessment of participant characteristics

Baseline characteristics (height, weight, age, gender, physical activity levels, and yoga experience) were measured or self-reported at the beginning of the participants' first in-person visit.

**Height, weight, and body mass index (BMI).** Height was measured twice using a wall-mounted stadiometer and measured to the nearest 0.1 cm. Weight was also measured twice with a digital scale to the nearest 0.1 kg. BMI in kg/m$^2$ was calculated from the height and weight measurements as weight (in kg) divided by height (in m$^2$).

**Age, gender, physical activity, and yoga experience.** Age and gender were self-reported by the participants. Participants were also asked to self-report how many minutes a week they were physically active, the other physical activities they participated in weekly, and the number of times a week they were currently practicing yoga.

## Experimental sessions

This study was a randomized crossover design (Fig 1) where the participants acted as their own control. Participant randomization to each order of conditions was determined through the sealed envelope technique. Under both conditions, the subjects completed the consent signing/orientation and then were randomized to an experimental condition when they attended their first in-person laboratory visit. Both conditions, the vinyasa yoga and prolonged seated control, lasted for 60 minutes.

To maximize internal validity, the in-person laboratory visits were scheduled for the same time and day of the week, one week apart. BP, HR, and HRV measurements were taken following a 5-minute seated rest prior to each condition, and at 5- and 65-minutes post conditions. Room temperature was monitored and maintained at a temperature between 68–72˚F (20–22.2˚C) during all sessions. Participants were asked to wear comfortable clothes to both sessions to allow for ease of movement. They were also asked to refrain from nicotine and caffeine for six hours, eating for three hours, and intense exercise and alcohol consumption for 24 hours prior to scheduled visits.

Participants completed self-selected desk work for approximately 50 minutes as a washout period between the 5- and 65-minute post condition rest and measurements. They were instructed to choose desk work that they did not perceive as stressful (e.g., reading, work on a personal laptop) and complete similar tasks during both sessions to standardize the washout period between conditions.

**Vinyasa yoga session.** The same yoga mat and block were provided for the participants to use during each yoga session. A Polar heart rate monitor and BP cuff were applied to the subjects while instructions about the yoga session were explained to them. Then, subjects were instructed to sit quietly for 5 minutes, upright, in a chair, with their feet planted directly on the floor. Following this initial rest, resting BP, HR, and a 6-minute HRV measurement were measured and recorded. Only the BP cuff was removed following these resting measurements prior to the yoga condition. The Polar heart rate monitor was continuously worn by subjects during the session to ensure participant safety with HR being recorded in the last 10 seconds of every minute. Following these baseline measurements but prior to beginning the yoga condition, the subjects had the opportunity to review a list of the yoga poses that they would be performing that day. After they reviewed this list, they were allowed to ask for any pose demonstrations or clarifications before beginning the yoga protocol.

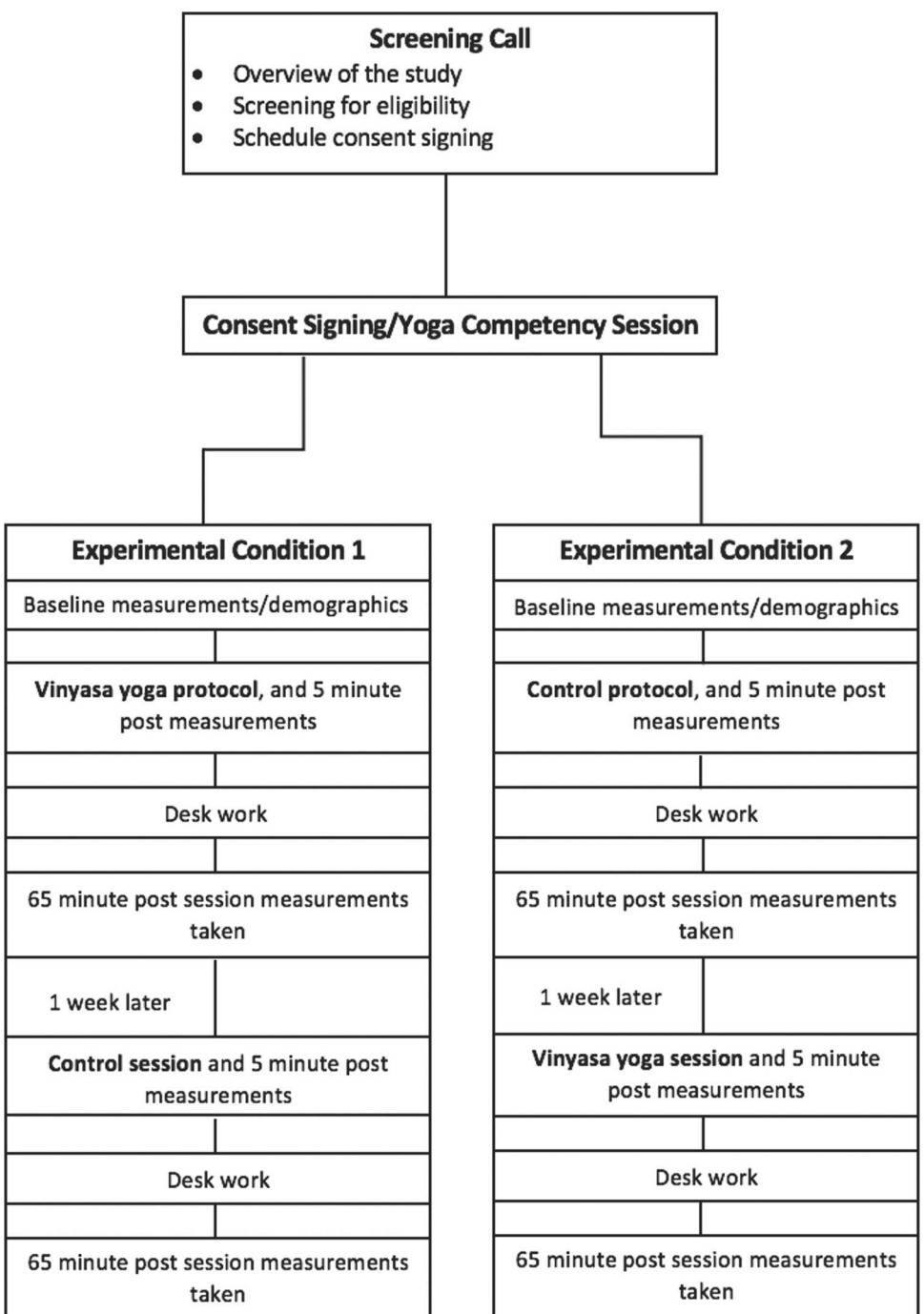

**Fig 1. Study design.** This figure displays the enrollment and study completion process for participants.

For the yoga condition, the assessor asked participants to follow a standardized yoga video that was used in the study conducted previously by our team [6]. This video was played on a large, 80-inch TV screen and included instructor cues of pose names and a person demonstrating the poses. For the current protocol (see S1 Fig), more instructor cues were added to this video providing additional breathing, and mindfulness cues. The yoga sequence was performed for 60-minutes and was a version of the *Journey into Power* sequence by Baron Baptiste

[16]. The subjects were allowed to take any pose modifications that they would typically use in their own yoga practice, and this information was recorded as a measure of fidelity. When the subjects were participating in the yoga condition, the investigator (a 200-hour certified yoga instructor) observed the subject's form and verbally corrected the subjects if it was necessary for safety concerns.

The BP cuff was re-placed on the subject when they moved to the resting measurement chair immediately following the yoga condition, and they rested again for 5 minutes. Following the 5-minute rest period, BP, HR, and HRV measurements were taken. The subjects relocated to a different chair in the same room where they performed seated desk work. At 60 minutes post yoga, subjects actively moved back to the resting measurement chair where they rested for a final 5 minutes and had their final measurements taken at 65 minutes post yoga.

**Prolonged seated control condition.** Before the prolonged seated control condition, the Polar heart rate monitor and BP cuff were applied to the subjects while the session instructions were explained to them. Then, subjects were instructed to sit quietly for 5 minutes, in a chair, in an upright position, with their feet planted directly on the floor. Following this 5-minute rest, subjects had their resting BP, HR, and a 6-minute HRV measurement taken. Similar to the yoga condition, the BP cuff was removed following these measurements and prior to the seated documentary viewing period. The Polar heart rate monitor remained on the subjects to record HR during the documentary viewing, in the last 10 seconds of every minute, to keep consistency between the conditions.

During the control condition, subjects moved from the measurement chair to be seated in a supportive chair within the same room. This chair is where they watched episodes of a nature documentary, *72 Cutest Animals*, [17] for 60 minutes. After the 60-minutes of documentary viewing, the subject moved back to the resting measurement chair and the BP cuff was re-placed. Following the same protocol as the yoga session, subjects rested for 5 minutes and had their BP, HR, and HRV measured. They moved from the measurement chair to perform seated desk work, returned to the measurement chair at 60 minutes post condition, completed the 5-minute rest, and had their other measurements taken beginning at 65 minutes post documentary viewing.

## Experimental session measurements

**Blood pressure.** BP was measured on the left arm with a Dinamap digital BP monitor with BP cuff size determined to ensure accurate measurements by finding the midpoint of the humerus and measuring the circumference at this midpoint. Both systolic BP (SBP) and diastolic BP (DBP) were recorded during BP measurements. BP was always taken following a 5-minute seated rest with the actual measurement occurring before, 5 minutes after, and 65 minutes after the vinyasa yoga and prolonged seated control conditions. The average from the two BP measurements was used for data analysis.

**Heart rate/heart rate variability.** HR and HRV were the final measurements taken at each measurement time point during the experimental sessions. They were measured for 6 minutes with a Polar H10 heart rate monitor. During resting conditions, the Polar H10 heart rate monitor is a highly validated for HR and HRV measures when compared to the electrocardiogram [18]. Kubios HRV software was utilized to analyze HRV data using R-R intervals including the time domain measures of mean HR, standard deviation of normal-to normal R-R intervals (SDNN), and root mean square of successive differences (RMSSD) [19]. Frequency domain variables included high frequency (HF) HRV and low frequency to high frequency ratio (LF/HF ratio). HR/HRV were measured before, 5-, and 65-minutes post conditions. Following the 5-minute rest and BP measurements, the HR/HRV recording lasted

6 minutes to record at least 5 minutes of usable data. Kubios also calculated breathing cycles per minute, so subjects were instructed to breathe normally during the 6-minute measurements.

### Sample size determination and statistical analyses

A power analysis was performed to determine sample size for this study. Though we did not find acute within-subject data that were appropriate to estimate an expected effect size, we did find evidence that other mind-body practices resulted in moderate effects on HRV [20]. Thus, we assumed a moderate effect size of 0.5, a within-subject correlation of 0.75, power of 0.80, and a type 1 error rate set at α = 0.05, which resulted in a required sample size of 18 participants.

Statistical Analyses were performed with Stata SE version 17.0 with a statistical significance level of p< 0.05 [21]. Data was checked to ensure normal distributions before analyses were conducted, and log transformations were performed when appropriate. Linear mixed effects models evaluated within-subject changes for each variable (SBP, DBP, mean HR, SDNN, RMSSD, and HF) at each time point (5 minutes and 65 minutes post conditions). Further linear mixed effects models compared each variable between conditions overall and at each time point, with adjustment for baseline values.

## Results

Of 22 individuals interested in volunteering for this study, 19 were found to be eligible, and 18 consented and participated in the study (Fig 2). One subject's HR/HRV data was excluded from analysis due to technical issues. Participant characteristics are reported in Table 1.

### Blood pressure

Effects of the experimental conditions on BP are reported in Fig 3. Within the vinyasa yoga condition, SBP and DBP were lower 5 minutes post yoga compared to baseline values (SBP: difference = -3.5 mmHg, p = 0.044; DBP: difference = -3.67 mmHg, p = 0.001). SBP was higher 5 minutes post seated control compared to baseline (difference = +4.8 mmHg, p = 0.040). Between conditions, SBP and DBP were lower 5 minutes post yoga compared to the seated control condition (SBP: difference = -8.14, p = 0.001; DBP: difference = -5.62, p = 0.001). There were no differences in BP either within or between conditions at 65-minutes post condition. Overall, BP was lower 5 minutes post yoga and was elevated following the seated control.

### Heart rate/heart rate variability

HR/HRV results are reported in Fig 4. Within the vinyasa yoga condition, HR was higher 5 minutes post yoga compared to baseline values (difference = +6.60 bpm, p = 0.010). Within the control condition, HR was lower at both post seated control time points compared to baseline values (5-minutes: difference = -4.58 bpm, p = 0.001); (65-minutes: difference = -2.20 bpm, p = 0.001). Between conditions, the change in HR from baseline was higher 5 minutes post (difference = +10.49, p = 0.001) and 65 minutes (difference = +4.70 bpm, p = 0.002) post yoga compared to the seated control. HR was not different at 65 minutes post yoga compared to baseline values.

For HRV outcomes, only RMSSD was lower 5 minutes post yoga compared to baseline values (difference = -0.28, p = 0.014), with no other significant differences. Significant increases were found within the seated control condition at both time points for SDNN, RMSSD, and HF (all p<0.01) but not LF/HF ratio. Between conditions, SDNN, RMSSD, and HF were lower

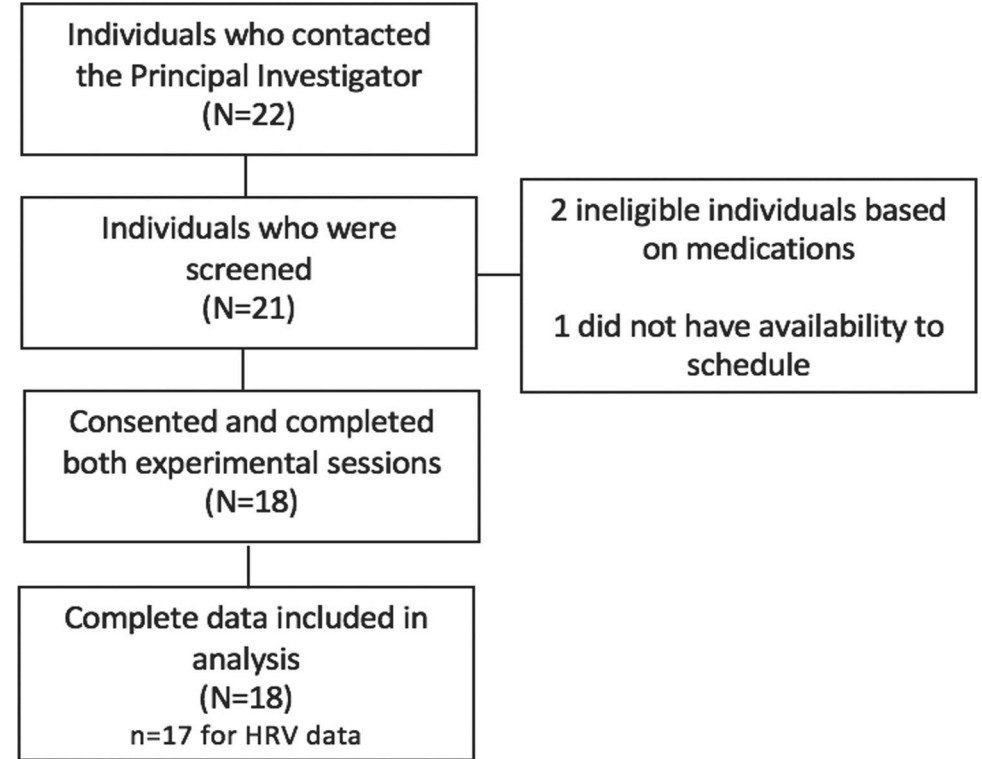

**Fig 2. Participant recruitment flow chart.** This flowchart displays participant recruitment and completion sample sizes.

post yoga compared to the seated control at both time points (all p<0.01). LF/HF ratio was higher in the yoga condition at 5 minutes post compared to the seated condition (difference = +0.38, p = 0.025) but not at 65 minutes post. Overall, HRV was mostly unchanged within subjects following yoga, though responses were significantly worse after yoga compared to the favorable responses observed following the seated control.

**Table 1. Participant characteristics.**

| Characteristics (N = 18) | Mean (SD) or n (%) |
|---|---|
| **Age** (years) | 28.7 (9.4) |
| **BMI** (kg/m$^2$) | 27.4 (5.6) |
| **Gender** | |
| Male | 9 (50) |
| Female | 9 (50) |
| **Weekly Exercise Minutes** | |
| 0–149 (min/week) | 3 (17) |
| >150 (min/week) | 15 (83) |
| **Number of Times Currently Practicing Yoga Weekly** | |
| 0 (days/week) | 11 (61) |
| >1 (days/week) | 7 (39) |

Data is presented as mean and standard deviation (SD) or sample size (n) and percentage (%)

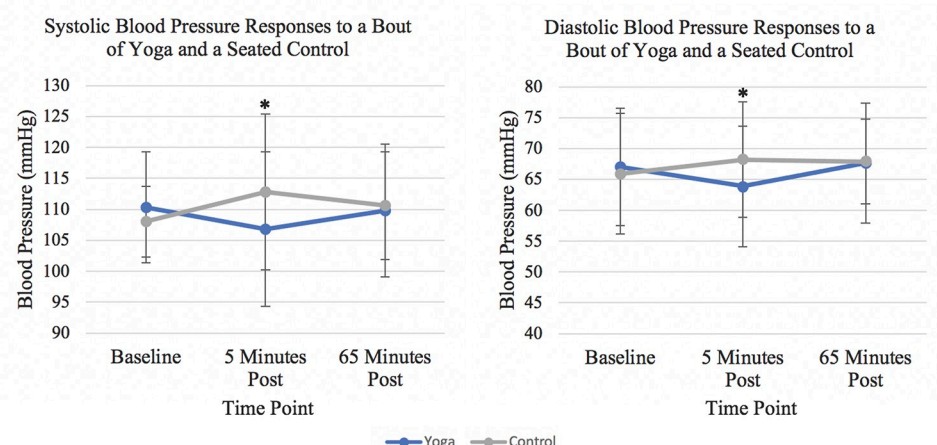

Fig 3. **Blood presure responses across conditions (N = 18).** This figure displays systolic and diastolic blood pressure responses in millimeters of mercury (mmHg) across the yoga and seated control condition; *indicates a statistically significant difference in the change from baseline between conditions; Abbreviations: SBP = Systolic Blood Pressure, DBP = Diastolic Blood Pressure.

| | Baseline-5 Minutes Post | | Baseline-65 Minutes Post | |
|---|---|---|---|---|
| **Within Yoga Condition** | **Mean Difference** | **p-value** | **Mean Difference** | **p-value** |
| **SBP (mmHg)** | -3.5 | *0.044 | -0.52 | 0.738 |
| **DBP (mmHg)** | -3.67 | *0.001 | +0.62 | 0.743 |
| **Within Control Condition** | | | | |
| **SBP (mmHg)** | +4.8 | *0.040 | +2.6 | 0.070 |
| **DBP (mmHg)** | +2.3 | 0.095 | +1.98 | 0.092 |
| **Between Conditions** | | | | |
| **SBP (mmHg)** | -8.14 | *0.001 | -2.76 | 0.136 |
| **DBP (mmHg)** | -5.62 | *0.001 | -0.70 | 0.668 |

## Discussion

This study examined physiological (SBP, HR, and HRV) responses to a vinyasa yoga session compared to a prolonged seated control session. Compared to the seated control condition, vinyasa yoga elicited the hypothesized and favorable BP response by decreasing both SBP and DBP. Contrary to the hypotheses, HR and HRV responses were unexpected compared to the control, with HR increasing, most HRV variables decreasing, and LF/HF ratio slightly increasing at both time points following the yoga condition compared to the control. It is worth noting that even though HR remained above baseline values by 65 minutes post yoga, HR returned clinically near baseline values by the final, 65-minute post measurement (+0.72 bpm). HRV was lower following the yoga condition compared to the control, but this is largely reflected by a significant increase in HRV during the seated control condition.

Post yoga BP responses have not been thoroughly studied with a rigorous study design, but our BP results are similar to a recent study; Pina et al., that found non-significant BP decreases following a 60-minute bout of vinyasa yoga compared to baseline values [22]. Our findings are also consistent with previously observed SBP responses to a single bout of exercise. Both aerobic [10, 23] and resistance [23] exercise bouts established acute post-exercise hypotensive BP responses. Finally, BP responses to this study's vinyasa yoga bout are consistent with a meta-analysis that found long term (4–52 week) yoga interventions generated moderate decreases in SBP and DBP, with greater reductions in BP from yoga interventions that included breathing awareness [24]. Proposed mechanisms for acute and chronic BP reductions post yoga are ANS influences, baroreceptor adjustments, and increased availability of vasodilators [9, 23]. These

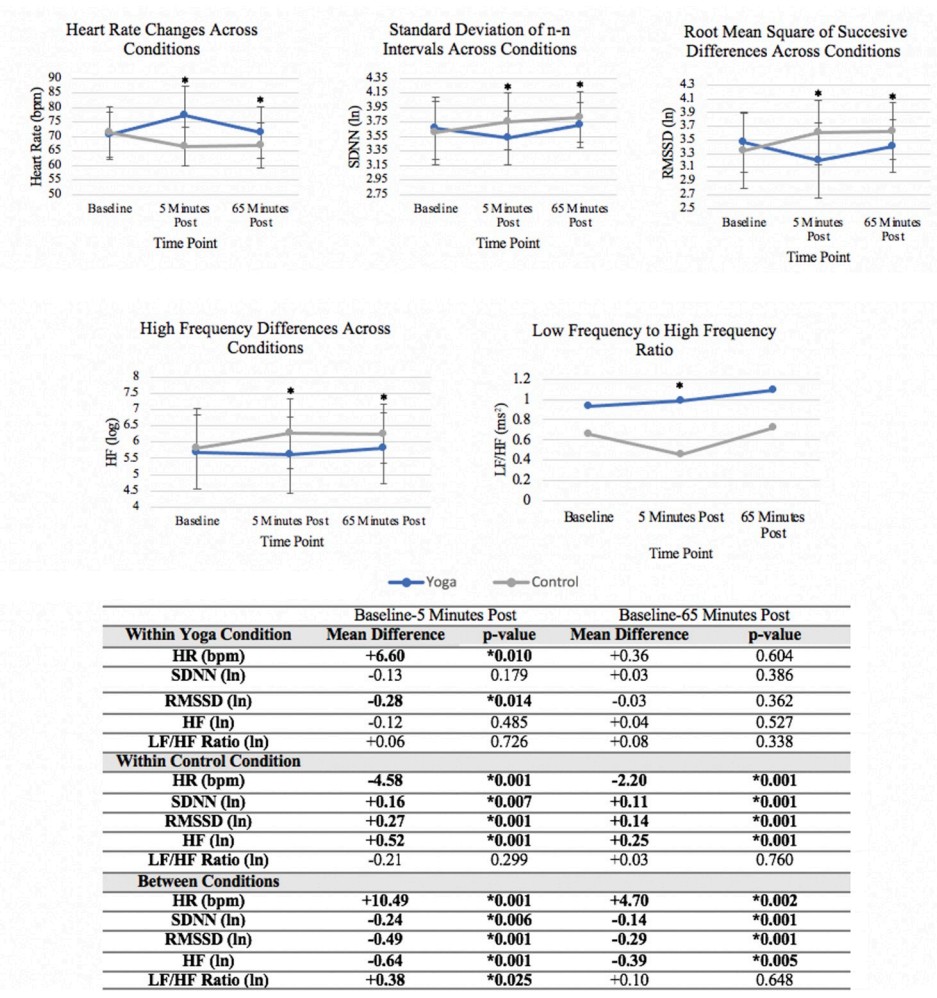

**Fig 4. Heart rate variability responses across the vinyasa yoga and prolonged seated control conditions (n = 17).** This figure displays heart rate variability responses across conditions; *indicates a statistically significant difference in the change from baseline between conditions; Abbreviations: SBP = Systolic Blood Pressure, DBP = Diastolic Blood Pressure, HR = Heart Rate, SDNN = Standard Deviation of Normal-to-Normal R-R Intervals, RMSSD = Root Mean Square of Successive Differences, HF = High Frequency, and LF/HF = Low Frequency to High Frequency Ratio.

conclusions and our study's findings are encouraging for utilizing vinyasa yoga as a method to improve BP, an aspect of cardiovascular health.

The BP responses to the seated control were also similar to previous studies. A study by our group found significant BP increases following a sedentary workday [25]. Lee et al., performed a systematic review and meta-analysis concluding that long-term sedentary behaviors are found to be associated with increases in both SBP and DBP [26]. These conclusions along with the findings of our study indicate that BP seems to increase in response to prolonged sitting, though importantly mechanisms explaining this effect are only partially understood. The proposed mechanisms for sitting-induced increases in BP include changes in ANS activity, [27] reduced nitric oxide availability, [28] reduced muscular demand, impaired vascular function, and reduced venous return in the lower limbs [29]. These events can lead to vasoconstriction [29] and fluid accumulation of the lower limbs [30], generating increases in BP. Future research is needed to clarify these mechanisms and identify strategies to reduce this effect.

Similar to post yoga BP responses, acute post yoga HR and HRV responses have not been thoroughly studied in large, diverse samples or utilizing standardized yoga protocols (i.e., reporting yoga style, yoga poses, pose duration, and breathing rate). Our HR and HRV responses post yoga were non-significant within subject and mostly unfavorable compared to the seated control. These HRV findings are consistent with a comprehensive review of some acute and long-term yoga interventions where mixed results were observed. As many studies in this review were poor quality with varying yoga intensities [31], our randomized crossover trial adds specific data about vinyasa yoga and acute effects. Furthermore, other studies investigating acute HRV responses to other types of exercise, such as aerobic and resistance training, have discovered that greater exercise intensity, duration, and volume more negatively influence autonomic responses, suggesting a dose-response relationship [11, 32, 33]. Further research is warranted with standardized and well-characterized yoga protocols to better understand autonomic responses to yoga.

The improved HR/HRV responses to the seated control from this study were not consistent with previous findings. Our group previously performed a meta-analysis of HR/HRV responses to bouts of prolonged sitting and found no significant changes in response to prolonged sitting bouts [34]. Although the reasons why our study's HR/HRV responses were different from those of the meta-analysis are unclear, the BP responses to our study's prolonged seated condition were typical. Taken together, the opposing cardiovascular responses to prolonged sitting, where BP increases, HR decreases, and HRV increases, may suggest that these changes may be strongly influenced by the different branches of the ANS acting independently of each other.

There are strengths and limitations to this study. The strengths include consistent visit times, the standardized yoga protocol, the randomized crossover design, an equal number of male and female subjects, measurement of respiration rate during HRV recordings, and a controlled room temperature. Despite these strengths, the limitations of our study should also be addressed. Weaknesses include utilizing a heart rate monitor rather than the gold standard electrocardiogram to measure HRV, studying a single style of yoga (vinyasa), including mostly participants that were meeting physical activity guidelines, a convenient (healthy) sample, and the laboratory setting which, though important for internal validity, may have reduced ecological validity. Further, though we intended our control condition to be a neutral comparison, prolonged sitting resulted in significant changes in our cardiovascular parameters which influence between-condition comparisons. The self-selected deskwork activity chosen for the washout period between post condition measures may also have influenced some of the changes observed in this study. Future studies should seek to recruit a more generalizable sample and investigate the effects of yoga over time in a parallel arm randomized controlled trial study design. Additionally, studies should consider including other yoga styles, other conditions that measure responses to the individual components of vinyasa yoga, comparison to a more neutral control condition or another type of moderate-intensity aerobic exercise, other ANS responses directly quantifying sympathetic activity, and follow-up measurements past 65 minutes following the cessation of exercise.

## Conclusion

The results of this study signify that vinyasa yoga alters some autonomic and cardiovascular functioning by decreasing SBP and DBP, increasing HR, and having minimal impacts on HRV. There were also opposing physiological effects in response to the prolonged seated control, and yoga resulted in improved BP but less favorable HR and HRV in comparison. The opposing cardiovascular responses in this study indicate that acute BP, as compared to HR

and HRV responses to vinyasa yoga and prolonged sitting may be predominantly influenced by different mechanisms. Overall, our findings potentially support vinyasa yoga as a strategy for acutely lowering BP though not for acute benefits in autonomic function.

## Supporting information

**S1 Fig. This figure displays the vinyasa yoga protocol followed for this study.** Poses are listed in order of how they were performed, and the number of breaths in each pose is listed after the pose name.
(TIFF)

**S1 Data. This is the minimal anonymized data set that was used for analyses in this manuscript.** The codebook is included as a second tab within the excel document.
(XLSX)

## Author Contributions

**Conceptualization:** Alexis Thrower, Bethany Barone Gibbs, Abdullah Alansare, Sally Sherman, Kelliann Davis.

**Data curation:** Alexis Thrower.

**Formal analysis:** Alexis Thrower, Bethany Barone Gibbs.

**Investigation:** Alexis Thrower.

**Methodology:** Alexis Thrower, Bethany Barone Gibbs, Abdullah Alansare, Sally Sherman, Kelliann Davis.

**Project administration:** Alexis Thrower.

**Resources:** Bethany Barone Gibbs.

**Software:** Alexis Thrower.

**Supervision:** Kelliann Davis.

**Visualization:** Alexis Thrower.

**Writing – original draft:** Alexis Thrower.

**Writing – review & editing:** Bethany Barone Gibbs, Abdullah Alansare, Sally Sherman, Kelliann Davis.

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
