## [Decision Letter · Decision Letter 0]

29 Aug 2023

PONE-D-23-23684Blood Pressure and Heart Rate Variability Responses to an Acute Bout of Vinyasa YogaPLOS ONE

Dear Dr. Thrower,

Thank you for submitting your manuscript to PLOS ONE. After careful consideration, we feel that it has merit but does not fully meet PLOS ONE’s publication criteria as it currently stands. Therefore, we invite you to submit a revised version of the manuscript that addresses the points raised during the review process.

ACADEMIC EDITOR:The study by Thrower et al. is a randomized crossover study that examines the effect of vinyasa yoga on autonomic nervous function, as measured by the changes in blood pressure, heart rate, and heart rate variability. The editor believes that the findings of your study will contribute to the current literature regarding the effect of mindfuleness-based interventions on autonomic nervous function. The manuscript is well-written and ready for revision.

We look forward to receiving your revised manuscript.

Kind regards,

Hidetaka Hamasaki

Academic Editor

PLOS ONE

Journal Requirements:

4. We notice that your supplementary [Figure S1] are included in the manuscript file. Please remove them and upload them with the file type 'Supporting Information'. Please ensure that each Supporting Information file has a legend listed in the manuscript after the references list.

Reviewers' comments:

Reviewer's Responses to Questions

**Comments to the Author**

1. Is the manuscript technically sound, and do the data support the conclusions?

Reviewer #1: Yes

Reviewer #2: Partly

2. Has the statistical analysis been performed appropriately and rigorously? 

Reviewer #1: Yes

Reviewer #2: Yes

3. Have the authors made all data underlying the findings in their manuscript fully available?

Reviewer #1: Yes

Reviewer #2: No

4. Is the manuscript presented in an intelligible fashion and written in standard English?

Reviewer #1: Yes

Reviewer #2: Yes

5. Review Comments to the Author

Reviewer #1: The manuscript is technically sound and presented lucidly . The data analysis and results are clear and understandably explained. The study protocol is clear. The discussion supports the study findings. However, the convenience sampling method along with the pre and post intervention pattern used for the study is a weaker option whereas a randomized control would have strengthened the study.

Reviewer #2: I reviewed with interest the article by Alexis Thrower and al. "Blood Pressure and Heart Rate Variability Responses to an Acute Bout of Vinyasa Yoga". In this article, the authors studied blood pressure and  heart rate variability response to an acute 60 minutes long bout of vinyasa Yoga compared to 60 minutes of prolonged seated condition when watching a relaxing television programme. They used a randomized crossover design. They were able to show that systolic and diastolic blood pressure significantly decreased immediately after vinyasa yoga when compared to prolonged seated condition. No such difference between the groups could be detected after 60 minutes of desk work. Heart rate was significantly higher, both immediately after vinyasa yoga and after 60 minutes of desk work, compared to prolonged seated condition. Heart rate variables lnSDNN and lnRMSSD were slightly, but statistically higher both immediately after Vinyasa yoga and after 60 minutes of desk work compared to prolonged seated condition. LnHF was slightly, but statistically higher immediately after vinyasa yoga compared to prolonged seated condition. However, when reviewing the manuscript, I had comments that I would like to receive answers from the authors. Some changes should be made by the authors as well in order to improve the article.

Firstly, the article's title does not reflect the study design concept. In the study design both interventions (vinyasa yoga and prolonged seated condition) are followed by 60 minutes of desk work. Type of desk work has not been specified and described in the article at all. Since the type of work and work itself has an effect on HRV variables this should be described in detail and mentioned/explained in the discussion. Example: Office workers with high effort-reward imbalance and overcommitment have greater decreases in heart rate variability over a 2-h working period. Jennifer L Garza et al. PMID: 25249418 DOI: 10.1007/s00420-014-0983-0. Heart rate variability and occupational stress—systematic review. Susanna JÄRVELIN-PASANEN et al. PMID: 29910218, doi: 10.2486/indhealth.2017-0190.

A title should be changed to reflect the true study design. Firstly there is vinyasa yoga intervention vs. prolonged seated intervention. First measurements were made right after that intervention, but after that a 60 minut desk work exposure follows. This could be seen as another important intervention by itself that affects later measurements and is not merely a time gap period for further evaluation of the yoga vs. prolonged seating. This should be further addressed in the Discussion as well.

Secondly, heart rate and heart rate variability measurements were made with the Polar heart rate monitor. No specific data on the model was provided and no possible validation studies regarding using that model were provided.Example: Validity of the Polar H10 Sensor for Heart Rate Variability Analysis during Resting State and Incremental Exercise in Recreational Men and Women. Marcelle Schaffarczyk. PMID: 36081005, doi: 10.3390/s22176536.This should be addressed and corrected by the authors.

Thirdly, references do not include previous studies involving yoga and HRV. Example: Yoga and heart rate variability: A comprehensive review of the literature. Anupama Tyagi. PMID: 27512317, doi: 10.4103/0973-6131.183712. This should be corrected and used in the Discussion section.

Fourthly, lnLF/HF ratio could also be used in HRV measurements. These results have not been shown in the paper. Authors should comment on that and/or include this data in the paper.

Authors should use corrections and additions to the paper related to all of the above mentioned points to further clarify the statement that different mechanisms are predominantly responsible for apparent discrepancy in the outcome variables (line 332). This should be addressed in somewhat greater detail in the discussion section, especially as proposed speculation/explanation for the observed discrepancy in (line 298 to 301) does not seem to make sense and no references were attached to this statement. Better speculations could be made if relying on the data already published. Examples: Cheema BS, Houridis A, Busch L, Raschke-Cheema V, Melville GW, Marshall PW, et al. Effect of an office worksite-based yoga program on heart rate variability: Outcomes of a randomized controlled trial. BMC Complement Altern Med. 2013;13:82. PMID: 23574691, 10. doi: 10.1186/1472-6882-13-82.Effective and Viable Mind-Body Stress Reduction in the Workplace:

A Randomized Controlled Trial. Ruth Q Wolever. PMID: 22352291 DOI: 10.1037/a0027278

Fourthly, Celsius degrees should be added to Fahrenheit degrees in order to further internationalize the paper. This should be corrected by the authors.

Lastly, conclusions should be based only on the data provided by the study. Data does not support the statement that vinyasa yoga improves cardiovascular health as HRV results are somewhat worse after yoga compared to prolonged sitting and significant reduction in blood pressure is only present immediately after yoga intervention. Authors should address this issue very carefully.

6. PLOS authors have the option to publish the peer review history of their article (what does this mean?). If published, this will include your full peer review and any attached files.

Reviewer #1: **Yes: **R.ARCHANA

Reviewer #2: **Yes: **Ivan Žebeljan

---

## [Author Response · Author response to Decision Letter 0]

27 Sep 2023

Dear Dr. Hamasaki,

Thank you for the opportunity to submit a revised draft of our manuscript now titled, ‘Blood pressure and heart rate variability responses following an acute bout of vinyasa yoga and a prolonged seated control: a randomized crossover trial’ to PLOS ONE. We appreciate you and the reviewers taking the time and effort to provide valuable feedback on this manuscript. For each comment, we have responded in a point-by-point fashion and incorporated changes to reflect the suggestions provided by the reviewers. All changes are detailed following the reviewers’ comments below and tracked within the manuscript Word document. We feel these edits have much improved the manuscript, and we hope you will too. We look forward to further review at PLOS ONE. 

Sincerely,

Alexis Thrower, MS

Graduate Student Researcher 

Doctoral Student in Exercise Physiology

Department of Exercise Physiology 

Comments from Reviewer 1 

Comment 1: The manuscript is technically sound and presented lucidly. The data analysis and results are clear and understandably explained. The study protocol is clear. The discussion supports the study findings. However, the convenience sampling method along with the pre and post intervention pattern used for the study is a weaker option whereas a randomized control would have strengthened the study.

Thank you for this suggestion regarding the sample and study design. We agree that more representative sampling and randomized clinical trial study designs with exposure to yoga and the observation of outcomes over time would have strengthened our study. We note that we did use a randomized crossover, within-subjects design with a comparison condition (not just pre- and post-) which is a strong design for acute effects. We have added the reviewer’s points as future directions to the last paragraph of the discussion section as follows (edits presented below with bold, blue font):

“The activity chosen for the washout period between post condition measures may also have influenced some of the changes observed in this study. Future studies should seek to recruit a more generalizable sample and investigate the effects of yoga over time in a parallel arm randomized controlled trial study design. Additionally, studies should consider including other yoga styles, other conditions that measure responses to the individual…”

Comments from Reviewer 2

Comment 1: Firstly, the article's title does not reflect the study design concept. In the study design both interventions (vinyasa yoga and prolonged seated condition) are followed by 60 minutes of desk work. Type of desk work has not been specified and described in the article at all. Since the type of work and work itself has an effect on HRV variables this should be described in detail and mentioned/explained in the discussion. Example: Office workers with high effort-reward imbalance and overcommitment have greater decreases in heart rate variability over a 2-h working period. Jennifer L Garza et al. PMID: 25249418 DOI: 10.1007/s00420-014-0983-0. Heart rate variability and occupational stress—systematic review. Susanna JÄRVELIN-PASANEN et al. PMID: 29910218, doi: 10.2486/indhealth.2017-0190. A title should be changed to reflect the true study design. Firstly there is vinyasa yoga intervention vs. prolonged seated intervention. First measurements were made right after that intervention, but after that a 60 minut desk work exposure follows. This could be seen as another important intervention by itself that affects later measurements and is not merely a time gap period for further evaluation of the yoga vs. prolonged seating. This should be further addressed in the Discussion as well.

We agree that the title could be more relevant to this manuscript and that we should include more information related to the desk work portion of the study. These changes can be found in the title on the title page of the document, in the short title of the header of each page, and are presented below in bold, blue font. In the methods section, changes were made to elaborate on what desk work included for these participants. A third paragraph was added to the Experimental sessions section and is included below for your reference. The limitation of the desk work activity was also addressed in the last paragraph of the discussion section and included below.

Title: Blood pressure and heart rate variability responses following an acute bout of vinyasa yoga and a prolonged seated control: a randomized crossover trial

Short title: Autonomic and cardiovascular responses to a yoga session and seated control

Methods: “Participants completed self-selected desk work for approximately 50 minutes as a washout period between the 5- and 65-minute post condition rest and measurements. They were instructed to choose desk work that they did not perceive as stressful (e.g., reading, work on a personal laptop) and complete similar tasks during both sessions to standardize the washout period between conditions.”

Discussion: “Further, though we intended our control condition to be a neutral comparison, prolonged sitting resulted in significant changes in our cardiovascular parameters which influence between-condition comparisons. The self-selected deskwork activity chosen for the washout period between post condition measures may also have influenced some of the changes observed in this study. Future studies should consider including other yoga styles, other conditions that measure responses to the individual components of vinyasa yoga, comparison to a more neutral control condition or another type of moderate-intensity aerobic exercise, other ANS responses directly quantifying sympathetic activity, and follow-up measurements past 65 minutes following the cessation of exercise.”

Comment 2: Secondly, heart rate and heart rate variability measurements were made with the Polar heart rate monitor. No specific data on the model was provided and no possible validation studies regarding using that model were provided.Example: Validity of the Polar H10 Sensor for Heart Rate Variability Analysis during Resting State and Incremental Exercise in Recreational Men and Women. Marcelle Schaffarczyk. PMID: 36081005, doi: 10.3390/s22176536.This should be addressed and corrected by the authors.

Thank you for bringing this to our attention. The model and a statement regarding the validity of the heart rate monitor was added under the Methods section within the first paragraph of the Heart rate/heart rate variability subheading. 

“HR and HRV were the final measurements taken at each measurement time point during the experimental sessions and using a Polar H10 heart rate monitor. During resting conditions, the Polar H10 heart rate monitor is highly validated for HR and HRV measures when compared to the electrocardiogram.(1) Kubios HRV software was utilized to analyze HRV data using R-R intervals including the time domain measures of mean HR, standard deviation of normal-to normal R-R intervals (SDNN), and root mean square of successive differences (RMSSD).(2)”

Comment 3: Thirdly, references do not include previous studies involving yoga and HRV. Example: Yoga and heart rate variability: A comprehensive review of the literature. Anupama Tyagi. PMID: 27512317, doi: 10.4103/0973-6131.183712. This should be corrected and used in the Discussion section.

Thank you for this suggestion. A statement and citation were added to the fourth paragraph of the discussion section to mention the comprehensive review of previous heart rate variability responses to yoga. 

“Similar to post yoga BP responses, acute post yoga HR and HRV responses have not been thoroughly studied in large, diverse samples or utilizing standardized yoga protocols (i.e., reporting yoga style, yoga poses, pose duration, and breathing rate). Our HR and HRV responses post yoga were non-significant within subject and mostly unfavorable compared to the seated control. These HRV findings are consistent with a comprehensive review of some acute and long-term yoga interventions where mixed results were observed. As many studies in this review were poor quality with varying yoga intensities,(3) our randomized crossover trial adds specific data about vinyasa yoga and acute effects. Furthermore, other studies investigating acute HRV responses to other types of exercise, such as aerobic and resistance training, have discovered that greater exercise intensity, duration, and volume more negatively influence autonomic responses, suggesting a dose-response relationship.(4–6) Further research is warranted with standardized and well-characterized yoga protocols to better understand autonomic responses to yoga.”

Comment 4: Fourthly, lnLF/HF ratio could also be used in HRV measurements. These results have not been shown in the paper. Authors should comment on that and/or include this data in the paper.

Thank you for suggesting this additional HRV variable. LF/HF ratio was added to this manuscript in the abstract, the methods section in the first paragraph of the Heart rate/heart rate variability subheading, and to the results section in the second paragraph under the Heart rate/heart rate variability section. Additionally, LF/HF ratio was added to Figure 4. 

Abstract: “The primary cardiovascular-related outcomes of this study included blood pressure (BP), heart rate (HR), and HR variability (HRV) measures [natural log transformed (ln) standard deviation of normal-to-normal R-R intervals (SDNN), root mean square of successive differences (RMSSD), high frequency (HF), and low frequency to high frequency ratio (LF/HF ratio)]. Linear mixed effects models were used for data analyses. Systolic BP was 8.14 mmHg lower at 5 minutes post yoga (p<0.001) but was not different 65 minutes post, compared to the control. HR was higher at 5- and 65-minutes post yoga compared to the control (10.49/4.70 bpm, respectively, both p<0.01). HRV was lower (worse) at 5 and 65 minutes post for lnSDNN, lnRMSSD, and lnHF (all p<0.01). LF/HF ratio was higher (worse) at 5 minutes post yoga compared to the control (difference= +0.38, p=0.025), but not different at 65 minutes post between conditions. Compared to prolonged sitting, vinyasa…”

Methods: “Kubios HRV software was utilized to analyze HRV data using R-R intervals including the time domain measures of mean HR, standard deviation of normal-to normal R-R intervals (SDNN), and root mean square of successive differences (RMSSD).(2) Frequency domain variables included high frequency (HF) HRV and low frequency to high frequency ratio (LF/HF ratio). HR/HRV were measured before, 5-, and 65-minutes post conditions. Following the 5-minute rest and BP measurements, the HR/HRV recording lasted 6 minutes to record at least 5 minutes of usable data.”

Results: “Significant increases were found within the seated control condition at both time points for SDNN, RMSSD, and HF (all p<0.01) but not LF/HF ratio. Between conditions, SDNN, RMSSD, and HF were lower post yoga compared to the seated control at both time points (all p<0.01). LF/HF ratio was higher in the yoga condition at 5 minutes post compared to the seated condition (difference= +0.38, p=0.025) but not at 65 minutes post. Overall, HRV was mostly unchanged within subjects following yoga, though responses were significantly worse after yoga compared to the favorable responses observed following the seated control.”

Discussion: “Compared to the seated control condition, vinyasa yoga elicited the hypothesized and favorable BP response by decreasing both SBP and DBP. Contrary to the hypotheses, HR and HRV responses were unexpected compared to the control, with HR increasing and most HRV variables worsening at both time points following the yoga condition compared to the control.”

Figure 4 addition: (included in revise and resubmit letter)

Comment 5: Authors should use corrections and additions to the paper related to all of the above mentioned points to further clarify the statement that different mechanisms are predominantly responsible for apparent discrepancy in the outcome variables (line 332). This should be addressed in somewhat greater detail in the discussion section, especially as proposed speculation/explanation for the observed discrepancy in (line 298 to 301) does not seem to make sense and no references were attached to this statement. Better speculations could be made if relying on the data already published. Examples: Cheema BS, Houridis A, Busch L, Raschke-Cheema V, Melville GW, Marshall PW, et al. Effect of an office worksite-based yoga program on heart rate variability: Outcomes of a randomized controlled trial. BMC Complement Altern Med. 2013;13:82. PMID: 23574691, 10. doi: 10.1186/1472-6882-13-82.Effective and Viable Mind-Body Stress Reduction in the Workplace:A Randomized Controlled Trial. Ruth Q Wolever. PMID: 22352291 DOI: 10.1037/a0027278

Thank you for the suggestions thus far and for requesting clarification regarding different mechanisms influencing the outcome variables. Adjustments were made in the second and fourth paragraph of the discussion section to clarify and simplify these paragraphs and this idea. We also made a small change to the conclusion paragraph. Your citation suggestions are included in the review that was added based on your third comment (above); therefore, we did not to add them individually in the discussion section. 

Second paragraph: “Finally, BP responses to this study’s vinyasa yoga bout are consistent with a meta-analysis that found longer term (4-52 week) yoga interventions generated moderate decreases in SBP and DBP, with greater reductions in BP from yoga interventions that included breathing awareness.(8) Proposed mechanisms for acute and chronic BP reductions from yoga are ANS influences, baroreceptor adjustments, and increased availability of vasodilators.(9,10) These conclusions and our study’s findings are encouraging for utilizing vinyasa yoga as a method to improve BP, an aspect of cardiovascular health.”

Fourth paragraph: “Similar to post yoga BP responses, acute post yoga HR and HRV responses have not been thoroughly studied in large, diverse samples or utilizing standardized yoga protocols (i.e., reporting yoga style, yoga poses, pose duration, and breathing rate). Our HR and HRV responses post yoga were non-significant within subject and mostly unfavorable compared to the seated control. These HRV findings are consistent with a comprehensive review of some acute and long-term yoga interventions where mixed results were observed. As many studies in this review were poor quality with varying yoga intensities,(3) our randomized crossover trial adds specific data about vinyasa yoga and acute effects. Furthermore, other studies investigating acute HRV responses to other types of exercise, such as aerobic and resistance training, have discovered that greater exercise intensity, duration, and volume more negatively influence autonomic responses, suggesting a dose-response relationship.(4–6) Further research is warranted with standardized and well-characterized yoga protocols to better understand autonomic responses to yoga.”

Conclusion: “There were also opposing physiological effects in response to the prolonged seated control, and yoga resulted in improved BP but less favorable HR and HRV in comparison. The opposing cardiovascular responses in this study indicate that acute BP as compared to HR and HRV responses to vinyasa yoga and prolonged sitting may be predominantly influenced by different mechanisms. Overall, our findings further support vinyasa yoga as a promising strategy for acutely lowering BP, but further investigation may be necessary to determine if vinyasa yoga improves other aspects of cardiovascular health.”

Comment 6: Fourthly, Celsius degrees should be added to Fahrenheit degrees in order to further internationalize the paper. This should be corrected by the authors.

Thank you for this suggestion. Celsius degrees were added to the methods section under the Experimental sessions subheading’s second paragraph and is displayed below with bolded and blue font. 

“Room temperature was monitored and maintained at a temperature between 68-72 °F (20-22.2 °C) during all sessions. Participants were asked to wear comfortable clothes to both sessions to allow for ease of movement. They were also asked to refrain from nicotine and caffeine for six hours, eating for three hours, and intense exercise and alcohol consumption for 24 hours prior to scheduled visits.”

Comment 7: Lastly, conclusions should be based only on the data provided by the study. Data does not support the statement that vinyasa yoga improves cardiovascular health as HRV results are somewhat worse after yoga compared to prolonged sitting and significant reduction in blood pressure is only present immediately after yoga intervention. Authors should address this issue very carefully.

Thank you for bringing this to our attention. We agree that our final conclusions could be tempered to more accurately reflect on our study findings. Adjustments were made to the last paragraph of the results section, the second paragraph of the discussion section, and the first paragraph of the conclusion section based off of this suggestion.

Abstract: “Compared to prolonged sitting, vinyasa yoga had variable effects on post-session autonomic function including favorable BP responses and unfavorable HR and HRV responses, further investigation is warranted.” 

Results: “Between conditions, SDNN, RMSSD, and HF were lower post yoga compared to the seated control at both time points (all p<0.01). LF/HF ratio was higher 5 minutes post between conditions (difference= +0.38, p=0.025) but not at 65 minutes post. Overall, HRV was mostly unchanged within subjects following yoga, though responses were significantly worse after yoga compared to the favorable responses observed following the seated control.”

Discussion: “Finally, BP responses to this study’s vinyasa yoga bout are consistent with a meta-analysis that found long term (4-52 week) yoga interventions generated moderate decreases in SBP and DBP, with greater reductions in BP from yoga interventions that included breathing awareness.(8) Previously suggested mechanisms for hypotensive BP responses to exercise are ANS influences, baroreceptor adjustments, and increased availability of vasodilators.(9,10) These conclusions and our study’s findings are encouraging for utilizing vinyasa yoga as a method to improve BP, an aspect of cardiovascular health.”

Conclusion: “The results of this study signify that vinyasa yoga alters some autonomic and cardiovascular functioning by decreasing SBP and DBP, increasing HR, and having minimal impacts on HRV. There were also opposing physiological effects in response to the prolonged seated control, and yoga resulted in improved BP but less favorable HR and HRV in comparison. The results of this study indicate that acute BP, HR, and HRV responses to vinyasa yoga and prolonged sitting may be predominantly influenced by different mechanisms. Overall, our findings potentially support vinyasa yoga as a strategy for acutely lowering BP though not for acute benefits in autonomic function.”

In addition to the above comments, additional edits were made to the formatting of the manuscript to better reflect the PLOS ONE guidelines and more information was added regarding the ethics of this study. Finally, all figures were uploaded to the Preflight Analysis and Conversion Engine (PACE) digital diagnostic and reuploaded to this submission. We look forward to hearing from you regarding this manuscript resubmission and to continue the peer review process.

---

## [Decision Letter · Decision Letter 1]

13 Nov 2023

Blood pressure and heart Rate variability responses following an acute bout of vinyasa yoga and a prolonged seated control: a randomized crossover trial

PONE-D-23-23684R1

Dear Dr. Thrower,

We’re pleased to inform you that your manuscript has been judged scientifically suitable for publication and will be formally accepted for publication once it meets all outstanding technical requirements.

Kind regards,

Hidetaka Hamasaki

Academic Editor

PLOS ONE

Additional Editor Comments (optional):

Reviewers' comments:

Reviewer's Responses to Questions

**Comments to the Author**

1. If the authors have adequately addressed your comments raised in a previous round of review and you feel that this manuscript is now acceptable for publication, you may indicate that here to bypass the “Comments to the Author” section, enter your conflict of interest statement in the “Confidential to Editor” section, and submit your "Accept" recommendation.

Reviewer #2: All comments have been addressed

2. Is the manuscript technically sound, and do the data support the conclusions?

Reviewer #2: Yes

3. Has the statistical analysis been performed appropriately and rigorously? 

Reviewer #2: Yes

4. Have the authors made all data underlying the findings in their manuscript fully available?

Reviewer #2: Yes

5. Is the manuscript presented in an intelligible fashion and written in standard English?

Reviewer #2: Yes

6. Review Comments to the Author

Reviewer #2: I reviewed with interest the updated article by Alexis Thrower and al. "Blood pressure and heart Rate variability responses following an acute bout of vinyasa yoga and a prolonged seated control: a randomized crossover trial". I feel that the authors have made changes that enable the article to be published.

7. PLOS authors have the option to publish the peer review history of their article (what does this mean?). If published, this will include your full peer review and any attached files.

Reviewer #2: **Yes: **Ivan Žebeljan

---

## [Editor Report · Acceptance letter]

16 Nov 2023

PONE-D-23-23684R1 

Blood pressure and heart Rate variability responses following an acute bout of vinyasa yoga and a prolonged seated control: a randomized crossover trial 

Dear Dr. Thrower:

I'm pleased to inform you that your manuscript has been deemed suitable for publication in PLOS ONE. Congratulations! Your manuscript is now with our production department. 

Kind regards, 

on behalf of

Dr. Hidetaka Hamasaki 

Academic Editor

PLOS ONE